# Stepwise Weighted Spike Coding for Deep Spiking Neural Networks

## Abstract

Spiking Neural Networks (SNNs) seek to mimic the spiking behavior of biological neurons and are expected to play a key role in the advancement of neural computing and artificial intelligence. The efficiency of SNNs is often determined by the neural coding schemes. Existing coding schemes either cause huge delays and energy consumption or necessitate intricate neuron models and training techniques. To address these issues, we propose a novel Stepwise Weighted Spike (SWS) coding scheme to enhance the encoding of information in spikes. This approach compresses the spikes by weighting the significance of the spike in each step of neural computation, achieving high performance and low energy consumption. A Ternary Self-Amplifying (TSA) neuron model with a silent period is proposed for supporting SWS-based computing, aimed at minimizing the residual error resulting from stepwise weighting in neural computation. Our experimental results show that the SWS coding scheme outperforms the existing neural coding schemes in very deep SNNs, and significantly reduces operations and latency.

## 1 Introduction

Spiking Neural Networks (SNNs) are known as the third generation of neural network models inspired by the biological structures and functions in the brain [32]. Unlike traditional Artificial Neural Networks (ANNs) that use continuous activation functions, SNNs incorporate discrete spiking events, enabling them to capture temporal dynamics and process information in a manner that closely mimics the brain's functioning [31]. This event-driven paradigm aligns with the brain's energy-efficient computation and has the potential for more efficient and lower-power computing systems. [33].

Various coding schemes have been proposed to describe neural activities, including rate coding and temporal coding [9]. Rate coding counts the number of spikes fired within a broad time window [23, 3, 18, 6], which effectively mitigates the impact of short-term interference on the signal. It was widely accepted in the early days and typically outperformed temporal coding [11, 34, 4, 29, 20]. However, the rate coding scheme disregards the information in the temporal domain of the input spike sequence and requires many pulses to represent the input signal value, making it an inefficient coding method that negates the low-power benefits of SNN. Due to the functional similarity to the biological neural network, spiking neural networks can embrace the sparsity found in biology and are highly compatible with temporal coding [31, 33, 27, 28, 21, 15]. Temporal coding relies on the specific timing or patterns of input spikes, allowing for greater information capacity in a single pulse. However, it requires a large number of time steps to provide fine-grained timing, which increases inference latency. Its sensitivity to variations in spike timing also makes it more vulnerable to temporal jitter or delays [25, 24]. Additionally, decoding temporal-coded information usually requires more complex neuron models [30, 36] and training methodologies [17, 26].

Submitted to 38th Conference on Neural Information Processing Systems (NeurIPS 2024). Do not distribute.

In the study of the temporal information dynamics of spikes, Kim et al. [16] discovered a phenomenon of temporal information concentration in SNNs. It is found that after training, information becomes highly concentrated in the first few timesteps. Based on this observation, we hypothesize that, from the perspective of the postsynaptic neuron, the first arriving spikes contain more information and require stronger responses. Consequently, we propose a mechanism whereby the neuron augments its own membrane potential with a specific coefficient prior to processing the subsequent input. This enhancement serves to increase the importance of preceding pulses on neurons, which is why the spikes are designated as Stepwise Weighted Spikes (SWS). Nevertheless, the amplification of the membrane potential makes it difficult for neurons to reduce its value through traditional "soft reset" (i.e. subtracted by an amount equal to the firing threshold), which can result in residual errors after neuron firing. To address this issue, we make the membrane potential reduced by a magnitude exceeding the threshold after firing. As a result, the membrane potential has both positive and negative residual values, which will generate both positive and negative spikes. This neuron is designated as a Ternary Self-Amplifying (TSA) neuron. To further reduce the error caused by the weighting process, a silent period is incorporated into the TSA neuron, allowing it to receive more input information before firing. We perform the classification tasks with SWS-based SNN on MNIST, CIFAR10, and ImageNet. The results show that the SWS coding scheme can achieve better performance with much fewer coding and computing steps. Even in very deep SNN, SWS coding scheme still performs well and achieves similar accuracy to the ANN with the same structure. Our major contributions to this paper can be summarized as follows:

- We propose the SWS coding scheme, which enables easy implementation of SNNs with low energy consumption and high accuracy. The stepwise weighting process enhances the information-carrying capacity of the preceding pulses, greatly reducing the number of coding spikes. Negative pulses are introduced in SWS coding to ensure an accurate information transmission.
- A novel TSA neuron model is proposed. TSA neuron progressively weights the input by augmenting its residual membrane potential before receiving the subsequent spike. The introduction of negative residual membrane potential and negative thresholds enhances the accuracy of the model's output.
- A silent period is added to TSA neuron to markedly improve accuracy at minimal latency cost. By adjusting the silent period step and coding step, SWS-based SNNs can exhibit performance advantages in different aspects, improving the flexibility of applications.

## 2   Related work

SNNs use spike sequences to convey information, making the encoding of real data into pulses a crucial step. Currently, the mainstream schemes of neural coding are rate coding and temporal coding [9, 33, 32]. Rate coding represents different activities with the number of spikes emitted within a specific time window. Due to its simplicity, rate coding is commonly used in deep learning of SNNs. However, it distributes information uniformly across a large number of spikes, resulting in an inefficient transmission process that increases network latency and energy consumption. Numerous researchers have proposed solutions to optimize inference latency in rate coding. Han et al.[11] proposed a "soft reset" spiking neuron model that retains a residual membrane potential after firing to better mimic the ReLU functionality. They demonstrated near lossless ANN-SNN conversion by using 2-8 times fewer inference time steps. Still, a delay of thousands of steps is required in large datasets or deep networks. In [14], Hu et al. reduced the encode time steps by converting a quantized low-precision ANN to a rate-coded SNN. They also proposed a layer-wise fine-tuning mechanism to minimize the inference latency. However, their neuron model and the subsequent fine-tuning algorithm are relatively complex. Furthermore, in deeper neural networks such as ResNet56, a 1.5% drop in accuracy can be observed. The above rate encoding solutions are limited because they do not consider the significance of each spike.

In [15], Kim et al. proposed phase coding, which assigns different weights to spikes based on their time phase. However, the transmission amount of information is bounded by the global phase, which causes inefficiency in hidden layers, resulting in a latency of up to three thousand steps for a 32-layer network. Burst coding [21] attempts to overcome this issue by introducing burst spikes, which utilize Inter-Spike Interval (ISI). Burst spikes are capable of conveying more information quickly and accurately by inducing Post-Synaptic Potential (PSP) dramatically. Nevertheless, it is still deficient

Table 1: Common symbols and their meanings in this paper.

| Symbol | Meaning |
| --- | --- |
| $S_i^l(t)$ | The spike train fired by the $i^{th}$ neuron in the $l^{th}$ layer |
| $u_i^l(t)$ | The membrane potential of the $i^{th}$ neuron in the $l^{th}$ layer |
| $z_i^l(t)$ | The integrated inputs to the $i^{th}$ neuron in the $l^{th}$ layer |
| $V_{th}^l$ | The firing threshold of the neurons in the $l^{th}$ layer |
| $\theta^l$ | The amplitude of the spikes fired by the neurons in the $l^{th}$ layer |

in terms of latency and efficiency. Rueckauer and Liu [27] proposed an efficient temporal encoding scheme where the analog activation values of the ANN neurons are represented by the inverse Time-To-First-Spike (TTFS) in the SNN neurons. Their new spiking network model generates 7-10 times fewer pulses by utilizing temporal information carried by a single spike. However, as pointed out in [10], TTFS coding scheme incurs expensive memory access and computational overhead, which diminishes the benefit of reduced pulse count. Furthermore, TTFS necessitates a large number of time steps to differentiate between various time points, which also increases network latency. Han and Roy [10] proposed the Temporal-Switch-Coding (TSC) scheme, in which each input image pixel is represented by two spikes, and its intensity is proportional to the timing between the two pulses. Their results showed a reduction in energy expenditure. However, TSC coding requires a large number of time steps to provide distinguishable time intervals, rendering it an ineffective approach to addressing the issue of the long latency.

Overall, rate coding employs a large number of pulses to encode information, which results in a considerable energy overhead and inference delays. On the other hand, temporal coding allows for greater information capacity in a single spike, but this does not reduce the computing latency as a precise time point or period can be identified only with a sufficient number of time steps. Therefore, new neural coding schemes should be developed.

## 3 Stepwise weighted spike coding scheme

### 3.1 Stepwise weighting

The spike train $S_i^l(t)$ of the $i^{th}$ neuron in the $l^{th}$ layer can be expressed as follows:

$$S_i^l(t) = \sum_{t_i^{l,(f)} \in F_i^l} \theta^l \delta(t - t_i^{l,(f)}) \tag{1}$$

where $\delta(t)$ is the Dirac delta function, $\theta^l$ is the spike amplitude of the $l^{th}$ layer, which is usually set to the same value as the firing threshold. $f$ is the index of the spike in the sequence, and $F_i^l$ denotes a set of spike times which satisfies the firing condition:

$$t_i^{l,(f)} : u_i^l(t_i^{l,(f)}) \geq V_{th}^l \tag{2}$$

where $u_i^l(t)$ denotes the membrane potential and $V_{th}^l$ denotes the firing threshold of the neurons in the $l^{th}$ layer.

Our basic idea is to amplify the membrane potential before the receipt of the subsequent input, which amplifies and prolongs the impact of the preceding input spikes on membrane potential, emulating the phenomenon of information concentration identified in [16]. For clarity, the meanings of important symbols are provided in table 1. The action of a neuron in SWS-SNN can be described as follows:

$$u_j^l(t) = \beta u_j^l(t-1) + z_j^l(t) - S_j^l(t) \tag{3}$$

where $\beta$ is the amplification factor which should be greater than one, $z_j^l(t)$ denotes the PSP (i.e. integrated inputs):

$$z_j^l(t) = \sum_i \omega_{ij}^l S_i^{l-1}(t) + b_j^l \tag{4}$$

where $\omega_{ij}$ is the synaptic weight and $b_j^l$ is the bias. Begin with the initial value $u_j^l(0) = 0$ and iteratively apply eq. (3) for each subsequent value until $u_j^l(n)$ and substitute eq. (1) and eq. (4) into it,

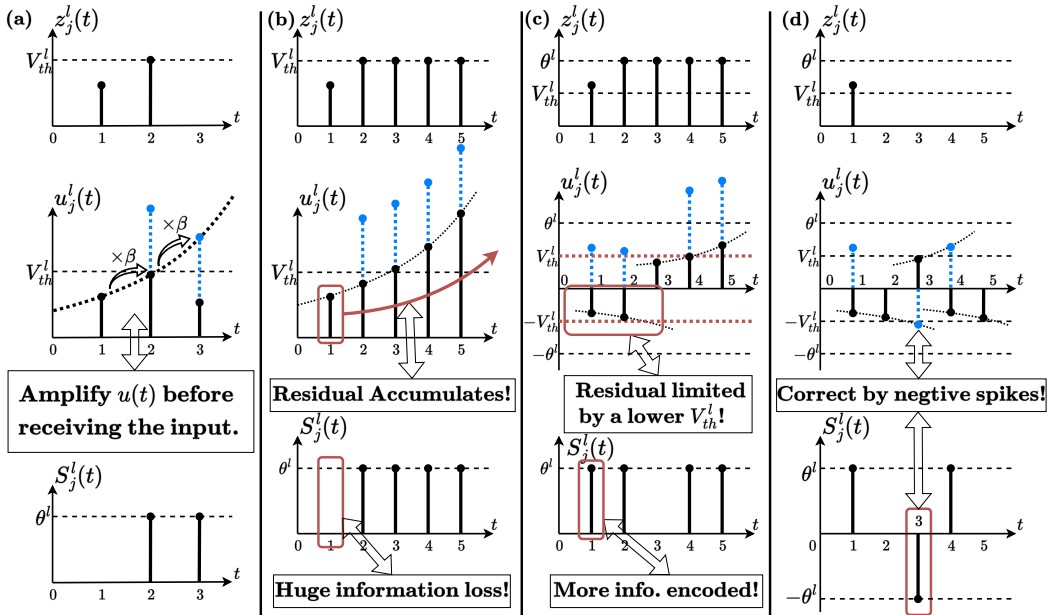

Figure 1: (a) Illustration of the stepwise weighting process. The meanings of the symbol $z_j^l(t)$, $u_j^l(t)$ and $S_j^l(t)$ can be found in table 1. The blue dotted line represents the membrane potential prior to the spike firing, and the black exponential function-like dotted line is employed to illustrate the trend of membrane potential amplification. (b) A $V_{th}^l$ equal to $\theta^l$ results in residual errors, leaving a lot of information unencoded. (c) $V_{th}^l$ is set to $\frac{1}{2}\theta^l$, which increases the possibility to fire spikes early to better limit the residual. (d) Use negative spikes to correct the excessively emitted information.

125  eq. (3) can be written as:

$$u_j^l(n) = \beta^n u_j^l(0) + \sum_{\tau=1}^{n} \beta^{n-\tau} z_j^l(\tau) = \sum_{t_i^{l-1,(f)}} \sum_i \sum_{\tau=1}^{n} \beta^{n-\tau} \omega_{ij}^l \theta^{l-1} \delta(\tau - t_i^{l-1,(f)}) + \beta^{n-\tau} b_j^l \quad (5)$$

126  Note that $S_j^l(t)$ is set to zero for simplicity. From eq. (5), it can be seen that the stepwise augment of
127  the membrane potential results in the spike input at time $t_i^{l-1,(f)}$ encoding the value $\theta^{l-1}\beta^{n-t_i^{l-1,(f)}}$.
128  This process is thus referred to as stepwise weighting, and $\beta^{n-t_i^{l-1,(f)}}$ serves as the weight. The
129  earlier the input pulse, the greater its ability to carry information. This solves the problem of excessive
130  encoding steps in previous schemes, allowing faster information transmission.

131  **3.2  Residual error**

132  Stepwise weighting effectively assigns more weight to earlier arriving pulses, but it also makes spike
133  generation more tricky. To ensure that input information is efficiently encoded and transmitted to
134  the next layer, the residual membrane potential should be minimized after neural computation is
135  completed. The stepwise weighting, however, amplifies the residual potential from the previous
136  time step. If $z_j^l(t)$ remains high in subsequent steps, reducing the membrane potential becomes
137  challenging, as shown in fig. 1(b). This vicious cycle ultimately leads to a persistently high membrane
138  potential, indicating that a substantial amount of information remains unencoded.

139  We refer to this phenomenon as residual error. One contributing factor is that the threshold is set
140  too high, resulting in a pulse being emitted only when the membrane potential exceeds the value $\theta^l$.
141  While this prevents excessive information transmission, it results in missed opportunities to bring
142  down $u_j^l(t)$ by firing a spike.

143  To address this issue, we propose setting the firing threshold $V_{th}^l$ to $\frac{1}{2}\theta^l$. This adjustment facilitates
144  pulse generation and reduces the residual membrane potential. After the neuron firing, the membrane
145  potential is subtracted by $\theta^l$, which leads to the emergence of a negative residual that will be stepwise

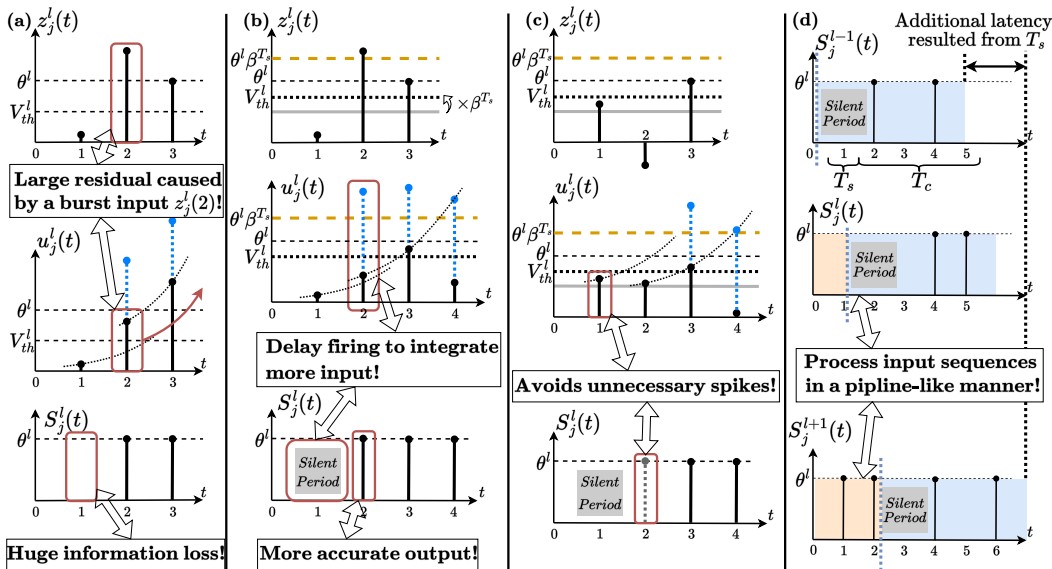

Figure 2: (a) Uncertainty in the input distribution leads to residual errors. (b) The silent period allows more information to be known when firing pulses. $T_s$ is set to 1 here. $V_{th}^l$ is amplified by $\beta^{T_s}$, and the original threshold is represented by a gray solid line. The orange dashed line represents the amount of membrane potential reduction after firing. (c) The silent period also avoids some unnecessary spikes and increases sparsity. Without the silent period, since $u_j^l(1)$ exceeds the original threshold, a pulse will be generated at $t = 1$, which will later be corrected by another negative spike. (d) The impact of the silent period on network latency. The output spike sequences corresponding to different inputs are drawn in blocks of different colors. The pulses drawn in the spike sequence are for illustrative purposes only.

weighted over time. The coefficient ½ is selected as it is capable of controlling both positive and negative residuals within a narrow and balanced range. A negative threshold $-V_{th}^l$ is introduced into the neuron model, which initiates a negative spike when the membrane potential falls below this threshold. This mechanism allows the excessively emitted information to be corrected by the negative spike, as shown in fig. 1(d). Given the above characteristics, we designate this neuron model as a TSA neuron.

## 3.3 Silent period

Another contributing factor to residual error is the imbalanced distribution of $z_j^l(t)$. A burst input of $z_j^l(t)$ at time point $\tau$ results in a sharp rise in membrane potential, making it difficult for subsequent spikes to reduce it, as shown in fig. 2(a).

This can be addressed by incorporating a silent period $T_s$ into the TSA neuron model. The neurons only integrates input and performs stepwise weighting, but are not allowed to fire in the first $T_s$ steps. This enables the acquisition of more known information before spike generation, resulting in increased accuracy, as illustrated in fig. 2(b). Since the preceding input information has been amplified by $\beta^{T_s}$ after the silent period, $V_{th}^l$ also needs to be adjusted accordingly, which is set to $\frac{\beta^{T_s}}{2}\theta^l$. Similarly, after firing, the membrane potential should be subtracted by $\theta^l \beta^{T_s}$. Note that the fired spike amplitude remains unchanged, that is, $\theta^l$.

The impact of the silent period on network latency is shown in fig. 2(d). The output results for different input sequences are distinguished by blocks of different colors. It can be observed that as network depth increases, the silent period accumulates, leading to a higher output latency. The inference latency of SWS-SNN can be calculated as follows:

$$T_{inf} = T_c + T_s \cdot L_{TSA} \tag{6}$$

where $T_{inf}$ is the inference delay, $T_c$ is the coding time steps, $T_s$ is the length of the silent period and $L_{TSA}$ is the number of TSA neuron layers. The neuron model in other coding schemes yields a zero $T_s$, leading to an output delay equal to the coding time step, which is consistent with the definition in the previous scheme. From fig. 2(d), it can be seen that different input sequences are processed in a pipeline-like manner, and the value of $T_c + T_s$ determines the throughput rate of SWS-SNN.

### 3.4 Input encoding

According to eq. (5), the value that can be losslessly encoded under the SWS coding scheme can be expressed as follows:

$$A_j = \sum_{\tau=1}^{T_c} a_j^\tau \cdot \theta^0 \beta^{T_c - \tau} \tag{7}$$

where $A_j$ denotes the encoded value. $a_j^\tau \in \{-1, 0, 1\}$ indicates the type of the output spike at time $\tau$: 1 for a positive pulse, $-1$ for a negative pulse and 0 for no pulse. $T_c$ denoted the time steps used for encoding. The weight $\beta^{T_c - \tau}$ results from the stepwise weighting process described in section 3.1. $\theta^0$ denotes the spike amplitude of the input encoding layer, which can be assigned an appropriate value based on the range to be encoded.

According to eq. (7), given a fixed $T_c$ and $\theta^0$, the distribution of $A_j$ is determined by $\beta$. Setting $\beta$ to 2 is reasonable, as it ensures $A_j$ is evenly distributed within the codable range. Compared to rate coding, which necessitates $2^{T_c}$ coding steps to encode the same range with same precision, SWS coding significantly enhances coding efficiency. Note that with the introduction of negative pulses, setting $\beta$ to 3 can also achieve a uniform distribution of $A_j$ and offers even more values for accurate encoding compared to $\beta = 2$.[1] When $\beta$ is less than 2, the distribution of $A_j$ becomes denser at smaller values, which may be suitable for encoding data that follows a similar distribution.

For static image classification tasks, the pixel value $p_j$ can be encoded by applying a constant input $z_j^0(t)$ to the TSA neuron. Considering the stepwise weighting process, we can write:

$$p_j = \sum_{\tau=1}^{T_c} |z_j^0| \beta^{T_c - \tau} \tag{8}$$

where $|z_j^0|$ denotes the amplitude of the constant input $z_j^0(t)$. Solve for $|z_j^0|$ and we have:

$$z_j^0(t) = \sum_{\sigma=1}^{T_c} \frac{p_j}{\sum_{\tau=1}^{T_c} \beta^{T_c - \tau}} \cdot \delta(t - \sigma) \tag{9}$$

Given that $z_j^0(t)$ is a constant at each step, $T_s$ can be set to 0 for the encoding layer. However, the neuron must await $T_s$ time steps after the completion of an encoding. This allows neurons in the subsequent layer to complete the previous neural computing before receiving the next encoded input.

## 4 Experiments

In this section, we convert quantized ANNs to SWS-based SNNs[2] and conduct experiments on MNIST, CIFAR10, and ImageNet. Firstly, an overview of SWS-SNN's performance across various datasets is provided. Subsequently, the network's inference latency and energy consumption is compared with other spike coding schemes. Finally, an ablation study is conducted to investigate the impact of lowered thresholds and silent periods on reducing residuals and enhancing accuracy.

ANNs used for conversion are all quantized to 8 bits. $\beta$ is set to 2 in the experiments to ensure that codable values are evenly distributed. Compared to $\beta = 3$, a smaller amplification factor reduces the impact of residual errors, resulting in more accurate output.

---

[1]Setting $\beta$ to 2 introduces some coding redundancy. E.g., $a_j^1 = 1, a_j^2 = -1$ and $a_j^1 = 0, a_j^2 = 1$ encodes the same amount of information.

[2]Details of the conversion process can be found in appendix A.1 and appendix A.2

Table 2: Performance on CIFAR10 and ImageNet.

| | Category | Methods | Architecture | Time Step | $T_s$ | SNN Acc | $\Delta$Acc[†] |
|---|---|---|---|---|---|---|---|
| CIFAR10 | Directly Learning | STBP-tdBN[35] | ResNet-19 | 6 | - | 93.16% | - |
| | | TET[5] | ResNet-19 | 6 | - | 94.50% | - |
| | ANN-SNN | TTRBR[20] | ResNet-18 | 64 | - | 95.04% | $-0.13\%$ |
| | | DSR[19] | PreAct-ResNet-18 | 20 | - | 95.24% | - |
| | | Calibration[18] | VGG-16 | 256 | - | 95.79% | $+0.05\%$ |
| | | OPI[1] | VGG-16 | 256 | - | 94.49% | $-0.08\%$ |
| | | Opt Conversion[4] | ResNet-20 | 128 | - | 93.56% | $+1.25\%$ |
| | ANN-SNN | **SWS (ours)** | ResNet-18 | 8 | 1 | 95.67% | $+0.22\%$ |
| | | | VGG-16 | 8 | 2 | 95.86% | $-0.04\%$ |
| ImageNet | Directly Learning | TET[5] | SEW-ResNet-34 | 4 | - | 68.00% | - |
| | | STBP-tdBN[35] | SEW-ResNet-34 | 4 | - | 67.04% | - |
| | | SEW Resnet[8] | SEW-ResNet-152 | 4 | - | 69.26% | - |
| | ANN-SNN | Hybrid training[26] | ResNet-34 | 250 | - | 61.48% | $-8.72\%$ |
| | | Spiking ResNet[13] | ResNet-50 | 350 | - | 72.75% | $-2.70\%$ |
| | | QCFS[2] | VGG-16 | 64 | - | 72.85% | $-1.44\%$ |
| | | Fast-SNN[14] | VGG-16 | 7 | - | 72.95% | $-0.41\%$ |
| | | COS[12] | ResNet-34 | 8 | - | 74.17% | $-0.05\%$ |
| | | RMP-SNN[11] | ResNet-34 | 4096 | - | 69.89% | $-0.75\%$ |
| | | TTRBR[20] | ResNet-50 | 512 | - | 75.04% | $-0.98\%$ |
| | ANN-SNN | **SWS (ours)** | VGG-16 | 8 | 2 | 75.27% | $-0.11\%$ |
| | | | ResNet-34 | 8 | 2 | 76.10% | $-0.08\%$ |
| | | | Inception-v3 | 8 | 2 | 76.70% | $-0.70\%$ |
| | | | ResNet-50 | 8 | 2 | 80.34% | $-0.35\%$ |
| | | | ResNeXt101_32x8d | 8 | 1 | 81.32% | $-1.17\%$ |
| | | | ResNeXt101_32x8d | 8 | 2 | 82.06% | $-0.42\%$ |

[†] $\Delta$Acc = Acc$_{\text{SNN}}$ − Acc$_{\text{ANN}}$

## 4.1 Overall performance

For simple classification tasks such as CIFAR10, our proposed SWS coding scheme has a faster inference speed than other ANN-SNN models while achieving similar classification accuracy, or has higher classification accuracy than direct learning at similar inference speeds. For example, ResNet18 with SWS improves throughput seven times over [20] while simultaneously improving accuracy. Although the network in [5] has a slightly higher throughput, its accuracy is 1.17% lower than our scheme. To fully test the potential of our proposed coding scheme, we conducted experiments on ImageNet using networks with various structures. The experimental results demonstrate that SWS coding has distinct advantages on extremely deep SNNs. Our SWS-based ResNet50 and ResNeXt101 achieved over 80% accuracy on ImageNet with only eight coding steps. The model in [12] achieves an almost lossless conversion with eight time steps. However, their method has to adjust the resting potential of neurons layer by layer, and the calibration effect for deeper networks is unclear. In [14], the original ANN needs to be quantized to 3 bits, resulting in a larger conversion loss. Directly trained SNNs typically achieve higher throughput, but their accuracy still requires improvement. In addition, the SWS coding scheme is easy to implement. No further fine-tuning is required after the conversion.

## 4.2 Accuracy vs. latency

The comparison of latency results between SWS-SNN and other ANN-converted SNNs[1, 11, 10, 4, 18, 2, 7] is illustrated in fig. 3. The latency of the network is calculated with eq. (6). In the counterpart models, the variation of delay is mainly caused by the changes in $T_c$. In contrast, $T_s$ determines latency in deep SWS-SNNs. Therefore, SWS-SNN has an upper limit on latency: $T_{inf}^{max} = T_c(1 + L_{TSA})$, which causes our curve to terminate earlier in fig. 3.

To ensure a fair comparison, we represent the ANN accuracy of each counterpart with dotted lines of the same color. The experimental results indicate that SWS-SNN can achieve optimal performance with minimal latency. Specifically, SWS-based VGG-16 can converge to the ANN performance

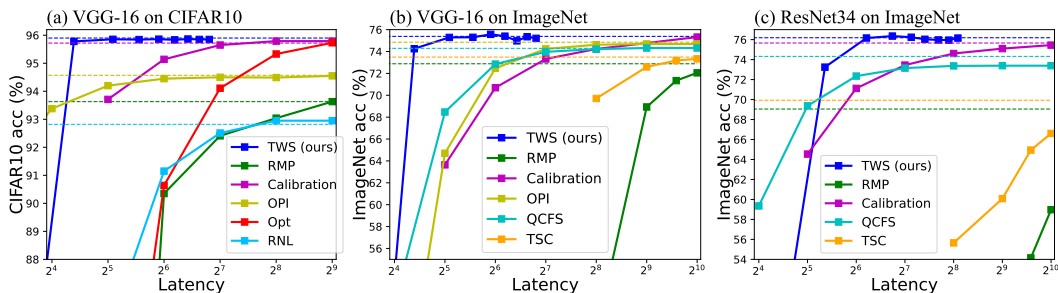

Figure 3: Latency versus accuracy. The ANN accuracy of each compared SNN is marked by dotted lines of the same colour. (a) VGG-16 on CIFAR10. (b) VGG-16 on ImageNet. (c) ResNet34 on ImageNet.

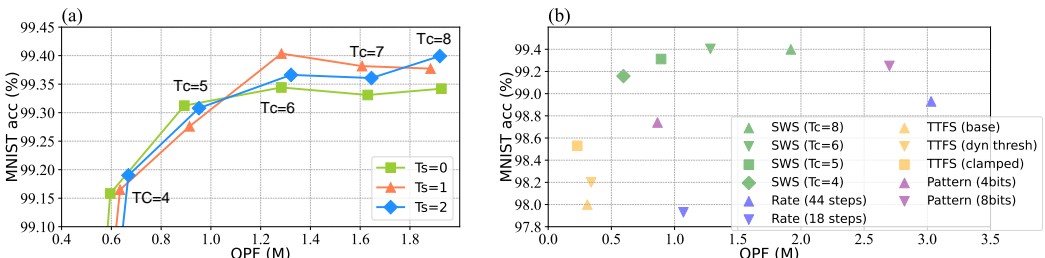

Figure 4: (a) Accuracy versus OPF with different combinations of $T_c$ and $T_s$. (b) Comparison of accuracy and energy consumption of SWS-SNN with other SNNs.

in the shortest time on CIFAR10 and reduce the inference latency on ImageNet by more than one order. Even though the silent period accumulates when the network gets deeper, the results in fig. 3(c) demonstrate that our scheme still achieves the fastest inference speed with the highest accuracy in a 34-layer network. Note that $T_s$ is set to the same value for each TSA layer for simplicity, resulting in discontinuous $T_{inf}$ values. This causes a sharp drop in accuracy at smaller delays.

## 4.3 Operation counting

To compare the energy consumption of SWS-SNN with SNNs under other encoding schemes, we adopt the method as in [29, 27, 28] to count operations:

$$OPF = (T_c + T_s)N_{TSA} + \sum_{l=1}^{L_{TSA}} \sum_{\tau=T_s l+1}^{T_s l+T_c} f_{out}^l n^l(\tau) \tag{10}$$

where $OPF$ (Operations Per Frame) denotes the number of operations for the classification of one frame, $T_c$ and $T_s$ denotes the coding steps and the length of the silent period, respectively. $L_{TSA}$ denotes the number of TSA layers, $f_{out}^l$ denotes the fan-out of neurons in layer $l$, $n^l(t)$ denotes the number of spikes fired in layer $l$ at time $\tau$ and $N_{TSA}$ denotes the number of TSA neurons. The first term on the right-hand side of the equation arises from the TSA's requirement to amplify the membrane potential. Note that due to the accumulation of $T_s$ over the network depth, the time period for counting $n^l(t)$ varies with $l$.

Experiments were conducted on MNIST using LeNet-5. We varied the silent periods and adjusted the coding steps to study their effects on OPF. The results are presented in fig. 4(a). As indicated in eq. (10), reducing $T_c$ lowers energy overhead. This presents a trade-off between energy consumption and inference accuracy, as fewer coding steps also reduce the number of values that can be accurately encoded. A larger $T_s$ requires TSA neurons to perform more operations to amplify the membrane potential. On the other hand, it reduces the number of unnecessary pulse emissions. Overall, silent period has a negligible impact on OPF.

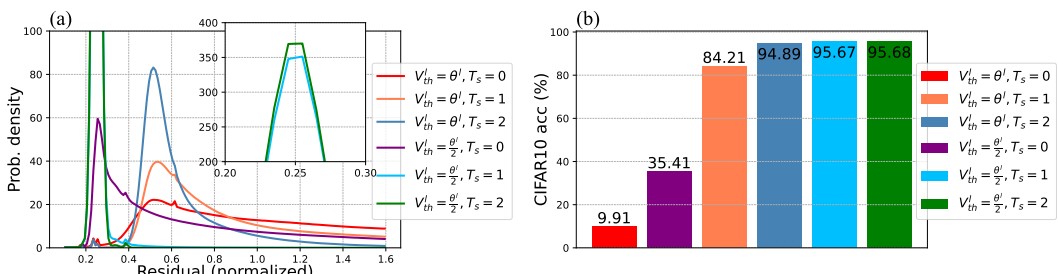

Figure 5: (a) The probability density of the residuals with/without a lowered $V_{th}^l$ and a silent period. (b) Inference accuracy of SWS-ResNet18 on CIFAR10 with/without a lowered $V_{th}^l$ and a silent period.

In fig. 4(b), the energy consumption of SWS-based SNN is compared with that of other SNNs. The experimental results demonstrate that our coding scheme can achieve a favorable balance between accuracy and energy consumption. The SWS coding scheme is superior to rate coding and temporal pattern coding in that it requires fewer operations and achieves higher accuracy. In TTFS encoding, each neuron fires at most one spike at a time, theoretically demanding the least OPF. With $T_c = 4$, SWS-SNN can achieve significantly higher accuracy with minimal increase in OPF. Note that if the ANN is quantized to a lower number of bits (e.g., 4 bits), the error caused by the reduced $T_c$ can actually be compensated by the quantization algorithm, which can potentially result in a higher performance.

## 4.4 Ablation study

In section 3.2 and section 3.3, we proposed reducing the firing threshold and introducing a silent period to mitigate residual error. To assess the impact of these two adjustments, we conducted experiments on CIFAR10 using ResNet18. After the neural computation, the residuals (absolute values) of the TSA neurons were analyzed. We first scaled the residuals by $1/\beta^{T_s}$ to counteract the effect of membrane potential amplification caused by the silent period, and then normalized them in units of $\theta^l$. The probability density of the residuals is shown in fig. 5(a).

The results demonstrate that lowering $V_{th}^l$ shifts the residual distribution from around $0.5\theta^l$ to approximately $0.25\theta^l$, corresponding to the quantization errors (i.e. rounding errors) under their respective thresholds. The addition of silent periods further concentrates the distribution and reduces large deviations. As can be seen from the green curve in fig. 5(a), setting $T_s$ to 2 and $V_{th}^l$ to $\theta^l/2$ makes the residuals almost all distributed around the quantization error. Compared to the red curve (without a lowered $V_{th}^l$ or a silent period), the residuals are greatly reduced, which fully proves the effectiveness of lowering the threshold and adding a silent period. The inference results on CIFAR10 is shown in fig. 5(b). When setting $V_{th}^l$ to $\theta^l$ and $T_s$ to zero, the network's output is almost random. Lowering the threshold and adding a silent period improve the accuracy to $35.41\%$ and $84.21\%$, respectively. Ultimately, the combination of both adjustments enabled SWS-ResNet18 to achieve an accuracy of $95.68\%$ on CIFAR10.

## 5 Conclusion

In this work, we have proposed a novel SWS spike coding scheme. The stepwise weighting process enhances the information-carrying capacity of the preceding pulses, greatly reducing the number of time steps for encoding. Combined with a silent period, our proposed TSA neuron model solves the problem of residual errors and achieves fast and accurate information transmission. Our experimental results have demonstrated that SWS coding is highly effective in extremely deep SNNs and achieves state-of-the-art accuracy. The SWS coding scheme is also highly flexible and can adapt to various needs.

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

# A Appendix

## A.1 Convert quantized ANNs to SWS-SNNs

A pretrained ANN was first obtained from torchvision, which is part of the PyTorch[22] project, and then quantized into $n$ bits following the Quantization-Aware Training (QAT) Workflow provided by PyTorch (8 bits in the actual experiment, with $n$ bits used here for generality). The quantized ANN can be characterized by the parameters listed in table 3, and the basic idea of the conversion process is illustrated in fig. 6(a). The activations of the quantized ANN can be mapped to an integer $Q$ between $[0, 2^n - 1]$ using a scaling factor $C$ and a zero point $Z$. With the same weight and bias between $Q_i^l$ and $Q_o^l$, the TSA layer can generate $S^l$, which encodes $Q_o^l$, provided that $S^{l-1}$ encodes $Q_i^l$ and no residual error occurs. In the actual SNN, the pulse amplitude $\theta^l$ is normalized to $1$. Therefore, the bias need to be further scaled to derive the final weight $W^l$ and bias $b^l$ for the SWS-SNN.

Table 3: The notations and meanings of parameters in the quantized network.

| Notation | Meaning |
|:---:|:---:|
| $\hat{X}_i^l$ | The quantized input of the $l^{th}$ layer |
| $\hat{X}_o^l$ | The quantized output of the $l^{th}$ layer |
| $C_i^l$ | The scaling factor of the quantized input of the $l^{th}$ layer |
| $Z_i^l$ | The zero point of the quantized input of the $l^{th}$ layer |
| $C_o^l$ | The scaling factor of the quantized output of the $l^{th}$ layer |
| $Z_o^l$ | The zero point of the quantized output of the $l^{th}$ layer |
| $\hat{W}^l$ | The quantized weight of layer $l$ |
| $C_w^l$ | The scaling factor of the quantized weight of layer $l$ |
| $Z_w^l$ | The zero point of the quantized weight of layer $l$ |
| $\hat{b}^l$ | The bias of layer $l$ |

The derivation is as follows. After QAT, we have:

$$\hat{W}^l \hat{X}_i^l + \hat{b}^l = \hat{X}_o^l, \tag{11}$$

$$Q_i^l = \frac{\hat{X}_i^l}{C_i^l} + Z_i^l, \tag{12}$$

$$Q_o^l = \frac{\hat{X}_o^l}{C_o^l} + Z_o^l, \tag{13}$$

where $Q_i^l$, $Q_o^l$ represent the integers to which the quantized input and output are mapped, respectively. Substitute eq. (12) and eq. (13) into eq. (11), and we can write:

$$\hat{W}^l (Q_i^l - Z_i^l) C_i^l + \hat{b}^l = (Q_o^l - Z_o^l) C_o^l, \tag{14}$$

which gives:

$$\begin{aligned} Q_o^l &= \hat{W}^l \frac{C_i^l}{C_o^l} Q_i^l + \frac{\hat{b}^l}{C_o^l} + Z_o^l - \frac{\hat{W}^l Z_i^l C_i^l}{C_o^l} \\ &= \tilde{W}^l Q_i^l + \tilde{b}^l, \end{aligned} \tag{15}$$

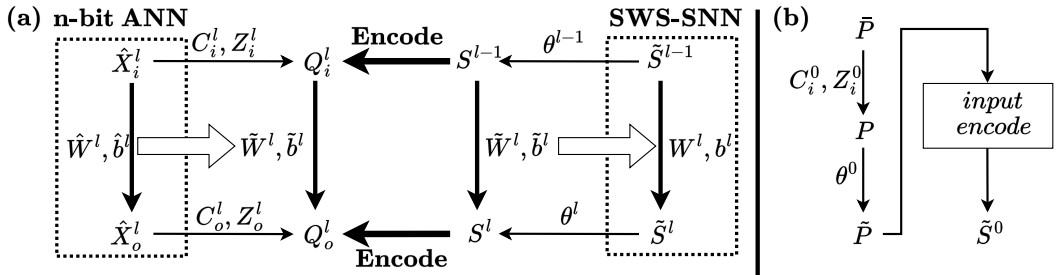

Figure 6: (a) Convert quantized ANNs to SWS-SNNs. $Q_i^l$ and $Q_o^l$ represent the integers to which $\hat{X}_i^l$ and $\hat{X}_o^l$ are mapped, respectively. $\tilde{W}^l$ and $\tilde{b}^l$ denotes the weight and bias to get $Q_o^l$ from $Q_i^l$. $W^l$ and $b^l$ denotes the weight and bias in SWS-SNN. The process of transferring weights and biases from the quantized ANN to SWS-SNN is indicated by white arrows. The core of the conversion is that the distribution of the integer $Q_o^l$ is known and can be easily encoded by $S^l$. (b) Process the input pixels to encode by pulses with an amplitude of $1$. $\bar{P}$ denotes the original pixel value, $P$ denotes the mapped value and $\tilde{P}$ denotes the value after scaled by $1/\theta^0$.

where

$$\tilde{W}^l = \hat{W}^l \frac{C_i^l}{C_o^l}, \tag{16}$$

$$\tilde{b}^l = \frac{\hat{b}^l}{C_o^l} + Z_o^l - \frac{\hat{W}^l Z_i^l C_i^l}{C_o^l}. \tag{17}$$

As seen in eq. (15), with the weight and bias set to $\tilde{W}^l$ and $\tilde{b}^l$ respectively, the layer outputs $Q_o^l$ when receiving $Q_i^l$. The pulse amplitude $\theta^l$ can be set to any value as long as the codable range calculated by eq. (7) covers $[0, 2^n - 1]$. Then we have:

$$W^l = \tilde{W}^l \frac{\theta^{l-1}}{\theta^l} = \hat{W}^l \frac{C_i^l}{C_o^l} \frac{\theta^{l-1}}{\theta^l} \tag{18}$$

Considering the membrane potential amplification, $b^l$ can be calculated as follows:

$$b^l = \frac{1}{\sum_{\tau=1}^{T_c} \beta^{T_c-\tau}} \tilde{b}^l = \frac{1}{\sum_{\tau=1}^{T_c} \beta^{T_c-\tau}} \left( \frac{\hat{b}^l}{C_o^l} + Z_o^l - \frac{\hat{W}^l Z_i^l C_i^l}{C_o^l} \right) \tag{19}$$

Once the $T_c$, $\beta$ and $\theta^l$ ($\theta^{l-1}$ is given by the previous layer) have been determined, all values on the right side of eq. (18) and eq. (19) are known. Consequently, $W_l$ and $b^l$ in the SWS-SNN can be readily calculated from the weight and bias of the quantized ANN.

After configuring the weights and biases as described above, the input pixel must be encoded into a pulse sequence with an amplitude of 1 as well. This process is illustrated in fig. 6(b). First, map the pixel value to $[0, 2^n - 1]$ using $C_i^0$ and $Z_i^0$ obtained from QAT. Assuming this range can be encoded by SWSs with an amplitude of $\theta^0$, scaling the pixel value by $1/\theta^0$ allows the use of a sequence with $\theta^0 = 1$ for encoding. Finally, encode the scaled pixels following section 3.4, and the required input spike sequence is obtained.

## A.2 Details for QAT

QAT is the quantization method that typically results in the highest accuracy. We basically follows the workflow provided by PyTorch. The default QAT quantization configuration is chosen to specify the kind of fake-quantization inserted after weights and activations. We choose Stochastic Gradient Descent (SGD) optimizer in QAT, with the value of momentum set to $0.9$ and the learning rate set to $1 \times 10^{-4}$ since the weights only need to be fine-tuned. QAT is done for 12 epochs and 20 batches in each epoch. We freeze the batch norm mean and variance estimates after three epochs and freeze the quantizer parameters (scaling factor and zero point) after another two epochs.

