# OpenReview forum: "Stepwise Weighted Spike Coding for Deep Spiking Neural Networks"
_NeurIPS.cc/2024/Conference — Submitted to NeurIPS 2024_

### Official Review · Reviewer_zR1q · 2024-07-05

**Soundness:** 2
**Presentation:** 2
**Contribution:** 2
**Rating:** 3
**Confidence:** 4

**Summary:**

This work belongs to ANN2SNN and proposes a novel coding scheme and neuron model to enhance the efficiency and accuracy of Spiking Neural Networks (SNNs) while reducing energy consumption. The Stepwise Weighted Spike (SWS) coding scheme improves information encoding by stepwise weighting input signals and introducing negative pulses, reducing the number of coding spikes needed. The Ternary Self-Amplifying (TSA) neuron model further enhances accuracy by progressively weighting the input through residual membrane potential adjustments and incorporating negative residuals and thresholds. Introducing silent periods allows the neuron to receive more input information before firing, significantly improving accuracy with minimal latency. Experimental results on datasets like MNIST, CIFAR10, and ImageNet demonstrate that the SWS coding scheme achieves better performance with fewer coding and computing steps, performing well even in very deep SNNs and achieving accuracy comparable to Artificial Neural Networks (ANNs) with the same structure.

**Strengths:**

Originality: This work introduces the Stepwise Weighted Spike (SWS) coding scheme, which is a novel approach in the field of Spiking Neural Networks (SNNs). The proposed method compresses spikes by weighting their significance in each step of neural computation, which enhances the performance and reduces the energy consumption of SNNs. Ternary pulses are a relatively new method in SNN, so the improvement of the ternary SNN encoding method has a relatively high degree of originality.

Quality: The paper provides a comprehensive set of experiments to validate the proposed SWS coding scheme. These experiments demonstrate that the SWS coding scheme significantly reduces operations and latency compared to existing neural coding schemes. The paper outlines the parameters used during training and provides justifications for the chosen experimental settings.

Clarity: The introduction of the paper effectively motivates the work by discussing the limitations of current SNN coding schemes and proposing SWS as a solution. The methodology is clearly presented, with detailed descriptions of the new coding scheme and the Ternary Self-Amplifying (TSA) neuron model. Important symbols and their meanings are well-explained, contributing to the overall clarity of the paper.

Significance: The paper makes a significant contribution by proposing the SWS coding scheme, which enhances the efficiency and performance of SNNs. This new method addresses critical issues such as high latency and energy consumption in existing coding schemes, making it a valuable addition to the field. By improving the encoding of information in spikes, the SWS scheme has the potential to advance the development of more efficient and lower-power computing systems, thereby providing new options for the choice of coding schemes in SNNs.

**Weaknesses:**

In the ImageNet experiments in Table 2, SWS and other comparative ANN-SNN methods used different baselines, which is why the '$SNN\  Acc$' results are much higher than those of the comparative methods. However, the ‘$\Delta ACC$’ does not seem to show a significant difference (except for Hybrid training and Spiking ResNet). Using the same network architecture and pre-trained weights would be more credible.

**Questions:**

1)

For the experiments on ImageNet in Table 2, it seems that the ANN baseline of SWS has a higher accuracy than that for the comparative methods. Why weren't other methods tested on the same architecture and pre-trained weights?

2)

Figure 2 describes the 'silent period' proposed in this paper, during which neuron potentials accumulate and are not allowed to fire spikes. After experiencing a silent period of $T_s$, the corresponding $\theta^l$ and $V_{th}^l$ are amplified by a factor of $\beta^{T_s}$. Firstly, should the $V_{th}^l$ in Figures 2(b) and 2(c) be $V_{th}^l\beta^{T_s}$? Otherwise, it does not correspond to $\theta^l\beta^{T_s}$. Secondly, how many time steps do the 'burst' spikes last after the silent period? Why, in the middle graph of Figure 2(b), are there larger spikes at the second and fourth time steps and a smaller spike at the third time step? In the middle graph of Figure 2(c), why is there a smaller spike at the third time step and a larger spike at the fourth time step? Is the amplification of $V_{th}^l$ and $\theta^l$ by $\beta^{T_s}$ maintained for several time steps or continuously after the silent period?

**Limitations:**

The authors have not explicitly addressed the limitations or potential negative societal impacts of their work. To improve the transparency and completeness of their research, the authors could consider the following constructive suggestions:

1)  Limitations:

Create a dedicated "Limitations" section in the paper to discuss any constraints, assumptions, or potential weaknesses of the proposed SWS coding scheme.

Reflect on the robustness of the results to violations of assumptions, such as noiseless settings, model specifications, or dataset dependencies.

Discuss the scope of the claims made in the paper, including the generalizability of the approach across different datasets and scenarios.
Address factors that may influence the performance of the SWS coding scheme, such as computational efficiency and scalability with varying dataset sizes.

Consider possible limitations related to privacy and fairness concerns in the implementation of the SWS coding scheme.

2)  Negative Societal Impact:

Explicitly acknowledge the potential negative societal impacts of the SWS coding scheme, such as privacy risks, fairness considerations, or unintended consequences.

Discuss how the technology could be misused or lead to harmful outcomes, even if not intended by the authors.

Consider mitigation strategies to address any identified negative societal impacts, such as controlled release of models, monitoring mechanisms, or additional safeguards.

Emphasize the importance of ethical considerations and responsible deployment of the SWS coding scheme in real-world applications.

---

> ### Author Rebuttal · Authors · 2024-08-07
>
> Thanks for your constructive and thoughtful comments. We are encouraged that you have recognized the novelty of our encoding scheme, the completeness of our experiments, the clarity of our presentation and the significance of our research. We would like to address your concerns and answer your questions in the following.
> > 1. For the experiments on ImageNet in Table 2, it seems that the ANN baseline of SWS has a higher accuracy than that for the comparative methods. Why weren't other methods tested on the same architecture and pre-trained weights?
>
> Thank you for pointing this out. Let us first explain the reason and then supplement the experiments. In conversion-based papers, almost every work uses different pretrained ANN weights. Considering the time cost of reproducing from scratch, we directly used the ANNs in the corresponding works as the accuracy baseline and added the conversion loss ($\Delta\text{Acc}$) as a standard. Since we used more recent pretrained ANN weights (most of the pretrained weights in this work are sourced from Torchvision or Hugging Face), our baseline might be higher.
>
> In the conversion-based ImageNet experiments in *Tab. 2*, [13] and [26] perform far worse than SWS-SNN in terms of both conversion loss and time steps; [20] and [11] fall far behind SWS-SNN in terms of time steps, and their conversion losses are also unsatisfactory. Compared to these four works, the performance advantage in our work is very clear. For [2] and [12], they actually require modifying the original ANN architecture and training from scratch. Specifically, they replace the ReLU in the original ANN with a QCFS function favorable for SNN conversion (the modified ANN is referred to as QCFS-ANN). Therefore, their ANN weights are necessarily different from the weights used in our work. In our opinion, this is unfair to SWS because QCFS-ANN is naturally more conducive to low time step conversion.
>
> For [7], we found it is open-source (the open-source works include: Hybrid Training [26], QCFS [2], Fast-SNN [14], and COS [12]), uses pretrained weights from Torchvision, and does not require structural changes. Therefore, we will supplement the experimental results using the same pretrained weights as [7], as shown in the table below.
>
> |ref    |architecture   |time step  |$T_{s}$    |SNN acc    |$\Delta$acc    |
> |:---:  |   :---:       |:---:      |:---:      |:---:      |:---:          |
> |[7]    |VGG-16         |$7$        |$-$        |$72.95$%  |$-0.41$%      |
> |ours   |VGG-16         |$8$        |$2$        |$73.28$%  |$-0.08$%      |
>
> We also want to highlight the difference in model structure. We are the only ones who conducted experiments on ResNeXt101, a network with 101 layers, which is a challenge for any encoding method. The results show that SWS encoding achieved a conversion loss of only $0.42$% with just $8$ coding steps, demonstrating the effectiveness of our approach.
>
> > 2. Figure 2 describes the 'silent period' proposed in this paper, during which neuron potentials accumulate and are not allowed to fire spikes. After experiencing a silent period of $T_{s}$, the corresponding $\theta^{l}$ and $V_{th}^{l}$ are amplified by a factor of $\beta^{T_{s}}$.Firstly, should the $V_{th}^{l}$ in Figures 2(b) and 2(c) be $V_{th}^{l}\beta^{T_{s}}$? Otherwise, it does not correspond to $\theta^{l}\beta^{T_{s}}$. Secondly, how many time steps do the 'burst' spikes last after the silent period? Why, in the middle graph of Figure 2(b), are there larger spikes at the second and fourth time steps and a smaller spike at the third time step? In the middle graph of Figure 2(c), why is there a smaller spike at the third time step and a larger spike at the fourth time step? Is the amplification of $V_{th}^{l}$ and $\theta^{l}$ by $\beta^{T_{s}}$ maintained for several time steps or continuously after the silent period?
>
> For the first question: While the firing threshold does increase by a factor of $\beta^{T_{s}}$, we still use $V_{th}^{l}$ to denote it because this aligns with the definition of the symbol. The inclusion of a silent period increases the reset magnitude of the membrane potential (i.e., $\theta^{l}\beta^{T_{s}}$) after firing a spike but does not change the amplitude of the spike (i.e., $\theta^{l}$) itself. Therefore, we use $V_{th}^{l}$, $\theta^{l}$, and $\theta^{l}\beta^{T_{s}}$ as symbols in this figure.
>
> For the second question: We apologize for the mistakes in *Fig. 2(b)* and *Fig.2(c)*. Your understanding is correct. The amplitude of all the blue dotted lines in the figure should be the same and equal to the amplitude indicated by the orange dashed line. We will correct this error in the full version of the paper. Thank you for pointing this out.

---

### Official Review · Reviewer_YWtV · 2024-07-06

**Soundness:** 3
**Presentation:** 2
**Contribution:** 2
**Rating:** 5
**Confidence:** 4

**Summary:**

This paper proposes a novel Stepwise Weighted Spike (SWS) coding scheme designed to improve the efficiency of Spiking Neural Networks (SNNs) by compressing spikes and weighting their significance in each step of neural computation. This method addresses the issues of high delays and energy consumption associated with existing SNN coding schemes, as well as the complexity of neuron models and training techniques. The authors also introduce a Ternary Self-Amplifying (TSA) neuron model, incorporating a silent period to support SWS-based computing. This model is designed to minimize the residual error resulting from the stepwise weighting process. The experimental results provided in the manuscript demonstrate that the proposed SWS coding scheme significantly outperforms existing neural coding schemes, particularly in very deep SNNs. Key improvements include reduced operations and latency, enhanced overall performance, and lower energy consumption.

**Strengths:**

1.	Innovative Approach: Introducing the SWS coding scheme and TSA neuron model is innovative.
2.	Performance Improvement: This paper provides experiments showing that the proposed methods outperform existing coding schemes regarding both performance and energy efficiency.

**Weaknesses:**

1.	Clarity and Detail: Some sections of this paper could benefit from more detailed explanations, particularly in the description of the SWS coding scheme and TSA neuron model. This would help in understanding the underlying mechanisms and their advantages.
2.	Comparative Analysis: While the experimental results are promising, there is no proof from the experimental results that the encoding method proposed is more advantageous.
3.	There are some grammatical errors in the paper. Such as the second paragraph of Section 3.3, "The neurons only integrates input and performs stepwise weighting". It is recommended that a uniform representation be used for "spike" and "pulse".
4.	Symbol design problem, "t" in Eq. (3) becomes "n" in Eq. (5).
5.	There are many long paragraphs and sentences in the paper, making it difficult for readers to accurately understand the meaning of the paper.
6.	The description of the problem in the third paragraph of Section 1 and the end of Section 2 is not clear, making it difficult for readers to understand the problem that the article really wants to solve.
7.	The description of the encoding method in Eq. (7) is difficult to understand. According to Eq.  (7), the encoded value $A_j$ should have no time step. However, in the experimental part, the method of this paper has 8 time steps.

**Questions:**

1.	The paper proposes a new encoding method. Is this encoding method designed for the TSA neuron model, or is it an encoding method in more general scenarios? If it is a targeted approach, would it be more appropriate as a sub-module of the proposed method?
2.	Three data sets are used in the experimental section. Why don’t show the comparison results on MNSIT in Table 2?

**Limitations:**

The paper mentioned that due to the setting of the neuron's silent period, the delay increases. It can be seen from the experiments that the overall latency of the method is lower, which can be regarded as solving this limitation. At the same time, this article does not have potential negative social impacts.

---

> ### Author Rebuttal · Authors · 2024-08-07
>
> Thanks for your constructive and valuable feedback. We are encouraged that you found our approach innovative and the performance satisfactory. We would like to address your concerns and your questions in the following.
>
> > 1. Clarity and Detail:
>
> Thank you for your constructive feedback. We apologize for the unclear descriptions in these sections. In the full version, we will provide a more detailed explanation of the residual error issue. We will also rewrite some of the long sentences for clarity.
>
> > 2. Comparative Analysis: While the experimental results are promising, there is no proof from the experimental results that the encoding method proposed is more advantageous.
>
> In our experiments, we have already compared accuracy, latency, and the number of operations with other encoding schemes. For instance, in *Tab. 2*, all SNNs using conversion strategies other than ours are based on rate encoding. It can be seen that most of them require very long time steps, and the conversion loss is not satisfactory (QCFS, COS, and Fast-SNN have shorter time steps because they have modified the original ANN to facilitate rate-based conversion). TTFS coding will also be added to the table in the full version (see our response to reviewer FCv3’s *question 4* for details).
>
> We compare the latency of SWS encoding with that of rate encoding and TSC encoding in *Sec. 4.2*. We did not find data on latency (measured by time steps) in the papers based on TTFS encoding, so it is not included in this part. In *Sec. 4.3*, we compare the operation numbers of SWS with other encoding schemes, including TTFS and rate encoding. These experimental results all show that SWS encoding has advantages.
>
> In the full version, we will be more explicit about the encoding scheme used in each compared work.
>
>
> > 3. There are some grammatical errors in the paper. Such as the second paragraph of Section 3.3, "The neurons only integrates input and performs stepwise weighting". It is recommended that a uniform representation be used for "spike" and "pulse".
>
> Thank you for pointing out these issues. We will check every detail carefully.
>
> > 4. Symbol design problem, "t" in Eq. (3) becomes "n" in Eq. (5).
>
> This is actually not a notation error. In *Eq. 5*, $n$ refers to a specific time point, whereas $t$ in *Eq. 3* is a variable.
>
> > 5. There are many long paragraphs and sentences in the paper, making it difficult for readers to accurately understand the meaning of the paper.
>
> Thank you for pointing out this issue. We will break down long sentences and appropriately shorten or split lengthy paragraphs.
>
> > 6. The description of the problem in the third paragraph of Section 1 and the end of Section 2 is not clear, making it difficult for readers to understand the problem that the article really wants to solve.
>
> In this paper, we tried to address the residual error problem caused by stepwise weighting. We will provide a clearer explanation of this issue and include additional mathematical definition to aid understanding (see our response to reviewer FCv3’s *question 2* for details).
>
> > 7. The description of the encoding method in Eq. (7) is difficult to understand. According to Eq. (7), the encoded value $A_{j}$ should have no time step. However, in the experimental part, the method of this paper has 8 time steps.
>
> For static image classification tasks, the encoding process is given by *Eq. 9* (not *Eq. 7*), where $p_{j}$ denotes the input pixel value and $z_{j}^{0}(t)$ denotes the input to the TSA neuron at each time step $t$. *Eq. 7* is used to specify the range of values that can be losslessly encoded by SWS encoding.
>
> > 8. The paper proposes a new encoding method. Is this encoding method designed for the TSA neuron model, or is it an encoding method in more general scenarios? If it is a targeted approach, would it be more appropriate as a sub-module of the proposed method?
>
> Yes, the SWS coding method is a targetted approach. According to *Eq. 5*, the SWS encoding scheme allows preceding spikes to encode more information. However, stepwise weighting can lead to an amplification of the residual membrane potential. The key to implement the SWS encoding is addressing the residual error. In this paper, we limit the value of the residual membrane potential by lowering the threshold and incorporating a silent period, resulting in the TSA neuron model. Therefore, our encoding scheme indeed requires specific neuron models to be effective, and our neurons are not interchangeable with those in other coding schemes.
>
> Thanks for this insightful comment. It would be more appropriate as a sub-module of the proposed method. We will put more emphasis on the neurons themselves and reorganize the content in the full version.
>
> > 9. Three data sets are used in the experimental section. Why don’t show the comparison results on MNSIT in Table 2?
>
> Thank you for pointing this out. We found that recent work seems to seldom compare accuracy on MNIST. Due to space constraints, we presented the experimental results on CIFAR10 and ImageNet, as we believe they are of higher priority. In our work, we use LeNet-5 to compare the number of operations on MNIST, and the comparison results can be found in *Fig. 4*. We have supplemented the experiment as requested, and the accuracy performance on MNIST is shown in the table below. We will include this in *Tab. 2* in the full version.
> |ref    |architecture   |time step  |$T_{s}$    |SNN acc    |$\Delta$acc    |
> |:---:  |   :---:       |:---:      |:---:      |:---:      |:---:          |
> |[6]    |LeNet-5        |$500$      |$-$        |$99.12$%  |$-0.02$%      |
> |[29]   |LeNet-5        |$44$       |$-$        |$98.93$%  |$-0.03$%      |
> |[27]   |LeNet-5        |$-$        |$-$        |$98.53$%  |$-0.43$%      |
> |ours   |LeNet-5        |$4$        |$0$        |$99.21$%  |$+0.13$%      |
> |ours   |LeNet-5        |$8$        |$0$        |$99.33$%  |$+0.25$%      |

---

### Official Review · Reviewer_FCv3 · 2024-07-08

**Soundness:** 2
**Presentation:** 2
**Contribution:** 2
**Rating:** 4
**Confidence:** 4

**Summary:**

The paper proposes a new coding scheme called Stepwise Weighted Spike (SWS) coding scheme for spiking neural networks to enhance the efficiency and reduce the number of operations and thus energy consumption. The SWS coding scheme tackles challenges associated with temporal and rate coding, such as heightened latency and energy usage. It achieves this by compressing spikes and assigning them varying weights at each computational step. Additionally, the paper introduces the Ternary Self-Amplifying (TSA) neuron model, which incorporates a silent phase to mitigate residual errors arising from the weighting procedure.

**Strengths:**

The SWS coding scheme enhances information capacity and reduces the number of spikes, leading to lower energy consumption and higher accuracy as compared to other coding schemes. The effectiveness of this approach is demonstrated using different datasets.

**Weaknesses:**

1. Which model of a spiking neuron is being employed in equation 3 (line 120)? What is the reset mechanism here after the neuron fires? Are the weights allowed to have negative values? The description of the model is unclear.

2. The notion of residual error intuitively makes sense but it is confusing. Please define the residual error mathematically (line 139) for better understanding.

3. Why ANN-(sws)SNN conversion is opted instead of directly training the SWS based SNN?

4. There are some recent works [1,2,3] with TTFS encoding which claims better results in regard to energy-efficiency and low-latency. First, these works need to be cited in the related work section. In my opinion, a detailed comparative analysis with other models and encoding schemes (for instance with [1,2,3]) needs to be carried out.

[1] Göltz, J., Kriener, L., Baumbach, A. et al. Fast and energy-efficient neuromorphic deep learning with first-spike times. Nat Mach Intell 3, 823–835 (2021).

[2] I. M. Comsa, K. Potempa, L. Versari, T. Fischbacher, A. Gesmundo and J. Alakuijala, "Temporal Coding in Spiking Neural Networks with Alpha Synaptic Function," ICASSP 2020 - 2020 IEEE International Conference on Acoustics, Speech and Signal Processing (ICASSP), Barcelona, Spain, 2020, pp. 8529-8533, doi: 10.1109/ICASSP40776.2020.9053856.

[3] Stanojević, Ana et al. “An Exact Mapping From ReLU Networks to Spiking Neural Networks.” Neural networks : the official journal of the International Neural Network Society 168 (2022): 74-88.

**Questions:**

1. Please address all the weaknesses above.

2. This is an empirical study. Can you provide theoretical proof that the number of time steps or operations required to interpolate a given set of data points (i.e., fitting the training data) is significantly less compared to other methods, such as the Time to First Spike, when using Equation 10 as a metric?

**Limitations:**

There is no potential negative societal impact and and one limitation related to the inclusion of silent period is noted in the main text.

---

> ### Author Rebuttal · Authors · 2024-08-07
>
> Thanks for your constructive and thoughtful comments. We are encouraged that you found our proposed coding scheme effective. We would like to address your concerns and answer your questions in the following.
>
> > 1. Which model of a spiking neuron is being employed in equation 3 (line 120)? What is the reset mechanism here after the neuron fires? Are the weights allowed to have negative values? The description of the model is unclear.
>
> For the first question: In *Eq. 3*, we aim to represent the process of stepwise amplification of the membrane potential. $z_{j}^{l}(t)$ is given by *Eq. 4*, and $S_{j}^{l}(t)$ is given by *Eq. 1* and *Eq. 2*. The spiking neuron in this equation can be simply understood as a variant of the basic IF neuron model, in which we use $\beta$ to represent the coefficient of the stepwise weighting.
>
> For the second question: This neuron adopts a soft reset mechanism, that is, the membrane potential is substracted by an amount equal to the firing threshold.
>
> For the third question: Yes, the weight can take a negative value.
>
> We apologize for being unclear in these parts of our paper, and we will be more explicit on all of these in the full version.
>
> > 2. The notion of residual error intuitively makes sense but it is confusing. Please define the residual error mathematically (line 139) for better understanding.
>
> Thank you for your valuable suggestion. Let us first explain the residual error in more detail: After the reset mechanism, there is always some residual membrane potential in the neuron (unless the membrane potential is exactly at the threshold before firing). This value can be used to measure the quality of the encoded information: We assume the time step for neural computation is $T$. In the best-case scenario, the residual membrane potential $u^{l}_{j}(T)$ is $0$. This means the input has been perfectly encoded and transmitted to the next layer, with no residual error. However, due to stepwise weighting, the residual membrane potential can easily accumulate over time (see *Fig. 1(b)*). The phenomenon of residual error refers to a situation where the residual membrane potential is significantly large (greater than the threshold) after neural computation finishes.
>
> The residual membrane potential $u_{j}^{l}(T)$ can be expressed by the following equation: $$u_{j}^{l}(T) = \sum_{\tau=1}^{T}\beta^{T-\tau}z_{j}^{l}(\tau)-\sum_{\tau=1}^{T}\beta^{T-\tau}S_{j}^{l}(\tau)$$ where $z_{j}^{l}(t)$ denotes the integrated input (see *Eq. 4*) and $S_{j}^{l}(t)$ denotes the output spike train (see *Eq. 1*). When $u_{j}^{l}(T)$ exceeds the firing threshold $V_{th}^{l}$, we refer to this as a residual error.
>
> > 3. Why ANN-(sws)SNN conversion is opted instead of directly training the SWS based SNN?
>
> Our experiments are based on conversion rather than direct training because the contribution of this paper lies in the encoding scheme. If we were to use direct training, any performance advantages might be attributed to new training algorithms. In our future work, we will incorporate training to demonstrate the benefits that it can bring.
>
> > 4. First, these works ([1,2,3]) need to be cited in the related work section. In my opinion, a detailed comparative analysis with other models and encoding schemes (for instance with [1,2,3]) needs to be carried out.
>
> Thank you for the references, which we will include in the full version of our work. In our experiments, we have already compared accuracy, latency, and the number of operations with other encoding schemes. For instance, in *Tab. 2*, all SNNs using conversion strategies (except ours) are based on rate coding. It can be seen that most of them require very long time steps, and the conversion loss is not satisfactory. QCFS, COS, and Fast-SNN have shorter time steps because they modified the original ANN to compensate for the shortcomings of rate encoding. QCFS and COS both replace ReLU in the ANN with the QCFS function to facilitate conversion, and Fast-SNN quantizes the ANN to only 2 bits.
>
> We will include *Ref. 3* you provided in *Tab. 2*, which achieves lossless conversion based on TTFS coding. However, *Ref. 3* does not explicitly provide the time steps used for their lossless conversion. Based on *Fig. 5* of that paper, we estimate that achieving lossless conversion for VGG-16 on CIFAR-10 requires approximately 150 time steps, which is much higher than the SWS coding scheme. Nonetheless, the high efficiency in terms of the number of operations with TTFS is undeniable, as we have compared in *Sec 4.3*.
>
> Additionally, in *Sec 4.2*, we compare the latency of SWS coding with rate coding and TSC coding. We did not find data on latency (measured in time steps) in papers based on TTFS encoding, so it is not included in this section.
>
> In the full version, we will be more explicit about the encoding scheme used in each compared work.
>
> > 5. Can you provide theoretical proof that the number of time steps or operations required to interpolate a given set of data points (i.e., fitting the training data) is significantly less compared to other methods, such as the Time to First Spike, when using Equation 10 as a metric?
>
> We believe that theoretically proving that SWS requires fewer operations per frame is not feasible. In *Eq. 10*, $n^{l}(\tau)$ represents the number of spikes fired by neurons at a specific time $\tau$, which is too complex to quantify. Therefore, in our experiments, we obtained the OPF number through statistical methods.
>
> However, it is feasible to make certain estimations based on *Eq. 10*. In *Sec. 3.4*, we analyzed the time steps required to encode the same range under SWS encoding and rate encoding, which are $T_{c}$ and $2^{T_{c}}$ respectively. Assuming that the neuron has a $50$% chance of firing a spike at each time step, then according to *Eq. 10*, the number of operations required for SWS encoding shows an exponential reduction compared to rate encoding.

---

> > ### Comment · Reviewer_FCv3 · 2024-08-10
> >
> > I appreciate the authors' responses, which have addressed most of my concerns. The proposal of a new encoding scheme is interesting; however, given that this is primarily an empirical study, further comparison with TTFS-based encoding schemes—both ANN-SNN conversion and direct training—is essential. Since TTFS encoding also emphasizes energy efficiency, low latency, and the number of operations based on optimal parameter training, a more thorough evaluation is necessary to truly assess the significance of the proposed approach. Based on this, I prefer to keep my rating unchanged.

---

### Official Review · Reviewer_ZNdM · 2024-07-12

**Soundness:** 3
**Presentation:** 3
**Contribution:** 2
**Rating:** 4
**Confidence:** 4

**Summary:**

The authors introduce a novel encoding method called Stepwise Weighted Spike (SWS) and a corresponding new neuron model named Ternary Self-Amplifying (TSA) for classification tasks utilizing the ANN2SNN training method. The proposed SWS encoding method assigns weights to the importance of spikes at each time step. The TSA neuron, which employs the SWS encoding method, features a lower threshold and includes a silent period.

**Strengths:**

1. Comprehensive method analysis: the authors conduct a thorough analysis of the Stepwise Weighted Spike (SWS) process, proposing a lower threshold and a silent period method to address residual error issues.

2. Superior Performance: the proposed method demonstrates superior performance in the field of ANN2SNN classification tasks.

**Weaknesses:**

1. Effectiveness of SWS: Various encoding methods, such as rate encoding and Time-to-First-Spike (TTFS) encoding, can be applied to different neurons and models. However, as illustrated in Figure 5, the SWS encoding method alone is ineffective without incorporating a lower threshold and a silent period. It only functions effectively when a neuron employs SWS encoding along with these additional components. Therefore, the paper should emphasize the neuron model rather than the encoding method, as it is not a universally applicable approach.
2. Lack of Experiments: The ablation study shows that the introduction of a silent period is the primary contributor to the improved performance. This raises doubts about the effectiveness of the SWS encoding method itself. Can the authors provide performance metrics for rate encoding combined with a lower threshold and silent period (if applicable) to ensure a fair comparison?

**Questions:**

See weaknesses.

**Limitations:**

See weaknesses.

---

> ### Author Rebuttal · Authors · 2024-08-07
>
> Thanks for your valuable and constructive feedback. We are delighted that you found our analysis of the method comprehensive and the experimental results satisfactory. We would like to address your concerns and answer your questions in the following.
>
> > 1. Effectiveness of SWS: Various encoding methods, such as rate encoding and Time-to-First-Spike (TTFS) encoding, can be applied to different neurons and models. However, as illustrated in Figure 5, the SWS encoding method alone is ineffective without incorporating a lower threshold and a silent period. It only functions effectively when a neuron employs SWS encoding along with these additional components. Therefore, the paper should emphasize the neuron model rather than the encoding method, as it is not a universally applicable approach.
>
>
> In fact, many works proposing new encoding schemes require the cooperation of corresponding neuron models, as seen in [1-3] below. There are many types of neurons available, and not all of them are suitable for existing encoding methods. Since our coding scheme is novel, and the meaning of each spike has changed, so we need to design new neurons.
>
> According to *Eq. 5*, the SWS encoding scheme allows preceding spikes to encode more information, demonstrating the effectiveness of this coding scheme to some extent. However, stepwise weighting can lead to an amplification of the residual membrane potential. The key to implement the SWS encoding is addressing the residual error. In this paper, we limit the value of the residual membrane potential by lowering the threshold and incorporating a silent period, resulting in the TSA neuron model. Therefore, our encoding scheme indeed requires specific neuron models to be effective, and our neurons are not interchangeable with those in other coding methods.
>
> Thanks for this insightful comment. We should put more emphasis on the neurons themselves. We will reorganize the content in the full version.
>
> [1] B. Rueckauer and S. -C. Liu, "Conversion of analog to spiking neural networks using sparse temporal coding," 2018 IEEE International Symposium on Circuits and Systems (ISCAS), Florence, Italy, 2018, pp. 1-5, doi: 10.1109/ISCAS.2018.8351295.
>
> [2] Han, B., Roy, K.: Deep spiking neural network: Energy efficiency through time based coding. In: Vedaldi, A., Bischof, H., Brox, T., Frahm, J.M. (eds.) Computer Vision – ECCV 2020. pp. 388–404. Springer International Publishing, Cham (2020)
>
> [3] S. Park et al., "Fast and efficient information transmission with burst spikes in deep spiking neural networks," in DAC, 2019.
>
> > 2. Lack of Experiments: The ablation study shows that the introduction of a silent period is the primary contributor to the improved performance. This raises doubts about the effectiveness of the SWS encoding method itself. Can the authors provide performance metrics for rate encoding combined with a lower threshold and silent period (if applicable) to ensure a fair comparison?
>
> Thanks for your valuable suggestion. Incorporating a silent period is a method to lower the residual membrane potential, which may also be effective for the basic IF model in rate coding. We have supplemented the experiment as you requested. The experiment was conducted on CIFAR10 using VGG-16, and the results are shown in the following table. $T_{s}$ is set to $32$ to ensure that it cannot be ignored compared to a large $T_{c}$. It can be seen the performance improvement brought by the silent period is not significant. This is because there is no stepwise weighting procedure in rate coding, and therefore, the residual membrane potential is not large. It is worth noting that in some cases (e.g., $T_{c}=128$), lowering the threshold or adding a silent period even leads to a decrease in performance.
>
> |Method|$V_{th}^{l}$|$T_{s}$|$T_{c}=64$|$T_{c}=128$|$T_{c}=256$|$T_{c}=512$|
> |:---:|:---:|:---:|:---:|:---:|:---:|:---:|
> |rate|$\theta^{l}$           |$0$    |$93.21$%|$95.33$%|$95.70$%|$95.80$%|
> |rate|$\frac{1}{2}\theta^{l}$|$0$    |$90.64$%|$94.89$%|$95.66$%|$95.84$%|
> |rate|$\theta^{l}$           |$32$    |$95.11$%|$95.17$%|$95.44$%|$95.70$%|
> |rate|$\frac{1}{2}\theta^{l}$|$32$    |$94.98$%|$94.64$%|$95.19$%|$95.66$%|
>
> We provide the accuracy of the original ANN and SWS-SNN ($T_{c}=8, T_{s}=1$) for comparison below. It can be seen that SWS-SNN brings improved accuracy and significant latency advantages (use *Eq. 6*). Note that the accuracy here differs from that in the paper because MaxPool was replaced with AvgPool in rate coding, necessitating the retraining of the ANN weights.
> |ANN|SWS-SNN ($T_{c}=8, T_{s}=1$) |
> |:---:|:---:|
> |$95.91\%$|$95.90\%$|

---

> ### Comment · Reviewer_ZNdM · 2024-08-13
>
> Thank you for addressing my concerns, I will keep the rating.

---

### Official Review · Reviewer_jeCw · 2024-07-14

**Soundness:** 3
**Presentation:** 2
**Contribution:** 2
**Rating:** 6
**Confidence:** 4

**Summary:**

The authors introduce a new spike coding scheme, which allows them to directly convert quantized ANN to their coding scheme. They demonstrate the effectiveness of their conversion on several pre-trained ANN with minimal loss in performance at the cost of an increase in latency.

**Strengths:**

- strong experimental results
- coding scheme appears to be novel

**Weaknesses:**

- limited connection to spiking neurons, a more straightforward motivation would be a temporal encoding of quantized ANN

**Questions:**

- could you compare your approach more explicitly to ref. 30?

**Limitations:**

- method only applicable to conversion from pre-trained ANN
- no demonstration of training of a model using this coding scheme.

---

> ### Author Rebuttal · Authors · 2024-08-07
>
> Thanks for your valuable comments. We are encouraged that you found our proposed coding scheme to have strong experimental performance. We would like to address your concerns and answer your questions in the following.
>
> > 1. Could you compare your approach more explicitly to *ref. 30*?
>
> We first discuss the similarities between the two papers. In *Ref. 30*, the authors proposed using the weighted sum of spikes to approximate the activation value in an ANN, aiming to reduce the number of spikes after ANN-SNN conversion. In our paper, *Eq. 5* indicates that stepwise amplification of the membrane potential results in spikes with different weights at different time steps. This highlights a similarity between the two approaches.
>
> When the generation time of a spike determines its weight, it becomes challenging for the neurons to fire both quickly and accurately. The problem arises due to the uncertainty of the input distribution in the temporal domain; neurons may receive a large amount of input after the moment for generating a high-weight pulse has passed. In *Ref. 30*, the authors addressed this issue by having neurons first receive all inputs and then generate spikes of different weights according to various thresholds. In our approach, we identified that the fundamental cause of this problem is the residual error resulting from membrane potential amplification. Consequently, our primary objective is to regulate the residual membrane potential. Initially, we attempted to lower the firing threshold and introduce negative pulses. However, this proved insufficient under certain extreme input conditions, leading us to implement a short silent period.
>
> From the results of the two methods, *Ref. 30* is equivalent to setting a silent period equal to the length of the coding time steps. Assuming the coding steps is $T_c$, *Ref. 30* requires $2T_c$ time steps to process an image. In contrast, SWS-SNN can achieve good results in $T_c+1$ time steps.
>
> From the perspective of motivation, the authors of *Ref. 30* directly aimed to reduce the number of spikes after ANN-SNN conversion. In contrast, we were inspired by the phenomenon of temporal information concentration and conceived the idea of amplifying the membrane potential at each time step. This led us to focus more on the behavior of spiking neurons, making it easier to identify residual errors and address the aforementioned issue more efficiently. In our opinion, our approach is more closely related to the spiking neurons.
>
> > 2. Method only applicable to conversion from pre-trained ANN & No demonstration of training of a model using this coding scheme.
>
> Thanks for your insightful comments. Our experiments are based on conversion rather than direct training because the contribution of this paper lies in the encoding scheme. If we were to use direct training, any performance advantages might be attributed to new training algorithms. In our future work, we will incorporate training to demonstrate the benefits that it can bring.

---

### Decision · Program_Chairs · 2024-09-25

**Decision:**

Reject

**Comment:**

This paper presents a new spike coding scheme. The scores are divided. After rebuttal and discussion, there are still some important concerns, such as the quality of the comprehensive set of experiments, that have not been handled, although this work may offer some interesting insights.